# Understanding the Relationship between Microstructure and Physicochemical Properties of Ultrafiltered Feta-Type Cheese Containing *Saturea bachtiarica* Leaf Extract

**DOI:** 10.3390/foods11121728

**Published:** 2022-06-13

**Authors:** Ali Alghooneh, Behrooz Alizadeh Behbahani, Maryam Taghdir, Mojtaba Sepandi, Sepideh Abbaszadeh

**Affiliations:** 1Health Research Center, Life Style Institute, Baqiyatallah University of Medical Sciences, Tehran P.O. Box 143591-3189, Iran; ali.alghooneh@gmail.com (A.A.); mtaghdir@gmail.com (M.T.); msepandi@gmail.com (M.S.); 2Department of Food Science and Technology, Faculty of Animal Science and Food Technology, Agricultural Sciences and Natural Resources University of Khuzestan, Mollasani P.O. Box 63417-73637, Iran; b.alizadeh@asnrukh.ac.ir; 3Department of Nutrition and Food Hygiene, Faculty of Health, Baqiyatallah University of Medical Sciences, Tehran P.O. Box 143591-3189, Iran

**Keywords:** extraction, microstructural, optimization, physicochemical, rheology, sensorial behavior

## Abstract

Microwave-assisted extraction was optimized to prepare *Satureja bachtiarica* leaf (SBL) extract based on antimicrobial (IZD) and antioxidant activities (DPPH) and extraction yield (EY). At optimum condition, i.e., 800 W power and 8 min, the best extraction results with EY = 16%, IZD = 73.56 mm, and DPPH = 24.2% were obtained. To develop a novel Feta-cheese, the influence of SBL extract, rennet, and starter concentrations were evaluated in terms of rheological, textural, and sensorial properties. At the optimized condition, the acceptance, taste, the strength of the network (*A*), and the distance between sequential cross-linking points (*ξ*) were 8.13, 8.07, 34,036.12 Pa·s^1/z^, and 5.41 nm, respectively. At the 60th day of storage time, the lowest *z* value (the network extensity parameter) of the cheese samples was observed. SEM image texture indices showed a good correlation with the studied instrumental texture parameters during 60 days of storage. The mold and yeast counts and their growth rate in the SBL extract-added cheese were lower than those for control one; whereas, the former cheese showed a greater LAB population between the 80th and 120th days. The antimicrobial and antioxidant qualities of SBL extract showed a significant influence on cheese properties.

## 1. Introduction

The growth in consumer demand for healthier food with lower artificial additives motivates the food industry and food scientists to develop functional foods with health-promoting bioactive ingredients. One of the primary ingredients in nutraceutical production is the herbal extract [1]. *Satureja bachtiarica* Bunge, a member of the *Lamiaceae* family with relatively wide distribution, has been cultivated as a vegetable plant for a long time. Recent research revealed that *Satureja* spp. essential oils have medicinal properties such as anti-inflammatory, antispasmodic, antidiarrheal, antioxidant, antiviral, antibacterial, and antifungal [2,3]. In a study by Dehkordi et al. [4], the *S. bachtiarica* ethanol extract showed an immunostimulant effect in Wistar rats. The high level of antioxidant and radical scavenging effect of the methanol extract from *S. bachtiarica* herb was explained by the high phenolic compounds (24.50 mg caffeic acid/g sample) and total flavonoids (87.99 mg catechin/g sample) [5]. In another study, the carvacrol and thymol phenolic compounds of *S. bachtiarica* essential oil as an antimicrobial agent were reported to be 49% and 4.5%, respectively [6]. Regarding *Satureja bachtiarica* leaf extract benefits, the application of this extract as an additive in one of the most consumed cheeses i.e., ultrafiltered Feta-type cheese (UF Feta-type cheese) could be promising. Plant extracts are obtained from dried plant materials by different extraction procedures such as microwave-assisted extraction [7,8]. Generally, microwave-assisted extraction is performed in a considerably shorter time and consumes less solvent as compared to conventional techniques [7]. UF Feta-type cheese is a white, soft, and spreadable cheese which is manufactured from ultrafiltered and pasteurized bovine milk mixed with mesophilic starter cultures and commercial microbial rennet [9]. Numerous studies on Feta-cheese had been performed to improve its sensorial, nutritional, and structural properties [10,11,12]. Even though Feta-cheese undergoes a heat treatment during production, it is susceptible to contamination by microorganisms, which results in a health risk for consumers and causes defects such as off-flavor, and changes in the texture and appearance. Concern about the safety of chemical preservatives has led to a growing interest in natural alternatives such as plant-based compounds. For example, Licón et al. [13] observed the antimicrobial effect of *Thymus vulgaris* essential oil on the growth of *Penicillium verrucosum* in ewes’ cheese. The addition of herb extract to cheese not only affects the microbiological properties of cheese but can also impact its physicochemical, textural, and sensorial behaviors. Rafiq and Ghosh [14] reported that the addition of fennel and ajwain extracts resulted in better overall acceptability scores, lower tyrosine and free fatty acid values, and lower values for the yeast and mold count of processed cheese at the end of shelf life.

To the best of our knowledge, no specific data are available regarding the optimization of SBL extract isolation and the extensive characterization of SBL extract-added UF Feta-type cheese from the mechanical, rheological, microstructural, and sensorial points of view. Therefore, the objectives of this study were: (i) to optimize the microwave-assisted extraction method, (ii) to formulate Feta-cheese using SBL extract, (iii) to evaluate the physicochemical, rheological, textural, microstructural, and sensorial properties of SBL-containing cheese within 60 days, and (iv) to compare the microbial properties of SBL-containing cheese with a control Feta-cheese in a 120-day storage period.

## 2. Materials and Methods

### 2.1. Materials

*Satureja bachtiarica* leaves (SBL) were purchased from a local store and their species were verified by the Institute of Medicinal Plants, Shiraz, Iran. The 2,2-Diphenyl-1-picrylhydrazyl hydrate (DPPH, 95% purity) and other chemicals of analytical grade were purchased from Sigma Chemical Co. (St. Louis, MO, USA). The starter (WhiteDaily 82) was obtained from Chr Hansen (Hoersholm, Denmark). The rennet (Fromase 2200 TL Granulate ≥ 2200 international milk-clotting units/g) was a microbial coagulant from DSM Food Specialties (Seclin, France). Salt (sodium chloride) was purchased from Merck (Darmstadt, Germany). Raw milk and cheese production equipment was provided by Amol Kalleh Dairy Co. (Mazandaran, Iran).

### 2.2. Satureja Bachtiarica Extract Isolation

First, the SBL was dried in an air forced oven at 38 °C. Then, dried leaves were powdered using a lab grinder (Waring) and stored at 4 ± 0.1 °C until the extraction time. Preliminary extractions were performed to find the best solvent mixture based on the highest yield, antimicrobial activity, and antioxidant activity in solvent maceration of SBL extract, which were 20.56 mL water, 175.52 mL ethanol, 48.51 mL methanol, and 5.41 mL glycerin. To enhance the extraction performance, the microwave technique was applied as the extraction aid. An amount of 50 g of powdered SBL was added to a mixture of solvents, i.e., 20.56 mL water, 175.52 mL ethanol, 48.51 mL methanol, and 5.41 mL glycerin, to make a constant 1:5 solid: solvent ratio. Microwave-assisted extraction was performed using the Tecnokit Chen (Italy, Tek-2611) microwave oven, at different operation times (5, 7.5, and 10 min) and power (400, 600, and 800 W). The extracts were filtered by Whatman filter paper under vacuum, followed by the centrifugation of filtrate in 3000× *g* for 10 min. Finally, the extracts were evaporated by a rotary vacuum evaporator at 40 °C, 60 rpm, and 200 mbar pressure until no solvent drips. The extracts were kept refrigerated (4 °C) in bottles until analysis.

### 2.3. Evaluation of Extraction Procedure

The extraction yield (EY) was determined from the percentage of concentrated extract weight divided by the weight of SBL powder. The antimicrobial activity of the SBL extract against Escherichia coli (*E. coli*) at 20 mg/mL concentration was evaluated using the disk diffusion procedure [15]. Furthermore, using the Brand-Williams et al. [16] method, the extract obtained from different procedures was tested against 2,2-diphenyl-1-picrylhydrazyl (DPPH) free radical to investigate their antioxidant capacity. 

### 2.4. Cheese Formulation

#### 2.4.1. Preparation of Feta-Cheese

Using the SBL extract, cheese samples were prepared according to the UF cheese making traditional method [17]. Briefly, milk was bactofugated, pasteurized (76 °C for 15 s), ultrafiltered, homogenized, and repasteurized (85 °C, 60 s). Then, the retentate was cooled to 34 °C when 1.4% salt and SBL extract (0.3, 0.6, and 0.9 g/100 kg cheese) were added. By adding the starter (1.5, 2.2, and 2.9 g/100 kg cheese), the pH of milk reached 6.2. Then, in the filler, rennet (1.5, 2, and 2.5 g/100 kg cheese) was mixed with water and added to each cheese container. Coagulation was performed at 34 °C for 25 min, then the container was sealed by aluminum foil. After dropping the pH to 4.85, cheese samples were cooled at 5 ± 1 °C for 3 days to complete the ripening time [9]. Subsequently, the rheological, textural, microstructural, and sensorial properties of these cheese samples were evaluated. 

#### 2.4.2. Characterization of Feta-Cheese

Strain sweep and frequency sweep experiments were performed according to Alghooneh et al. [18] using a Physica MCR 300 rheometer (Anton Paar, GmbH, Graz, Austria) with cone-plate geometry. The samples were held at room temperature for 1 h before the rheological measurements and their periphery was coated with light silicone oil to minimize loss of water. The limit of linear viscoelastic behavior of the Feta-cheese was defined at room temperature by applying 1 Hz frequency and 0.01–1000% strain. Subsequently, the frequency sweep measurement was performed over a frequency range of 0.0628–62.8 rad·s^−1^ within the LVE range (strain amplitude of 0.01%). The intercepts (*k*′ and *k*″) and slopes (*n*′ and *n*″) of frequency-dependence of storage and loss modulus, respectively; the overall loss tangent (*k*″/*k*′), the strength of the network (*A*), the network extensity parameter (*z*), the distance between sequential crosslinking points (*ξ*), the slope of double logarithmic scale plot of complex viscosity vs. frequency (*η*s*), and the relaxation time (*λ_irel_*) were determined with the following equations [18].
(1)$$G^{\prime }=k^{\prime }\times \omega^{n^{\prime }}$$
(2)$$G^{\prime \prime }=k^{\prime \prime }\times \omega^{n^{\prime \prime }}$$
(3)$$G^{\prime }=\sum_{i=1}^{n}\frac{G_{i}}{\left(1+\omega^{2}\times \lambda_{irel\;}^{2}\right)}$$
(4)$$G^{\prime \prime }=\sum_{i=1}^{n}\frac{G_{i}}{1+(\omega^{2}\times \lambda_{irel}^{2})}$$
(5)$$G^{*}=A_{a}\times \omega^{z}$$
(6)$$\upsilon =\frac{G_{p}^{\prime }N_{A}}{RT}$$
(7)$$\xi ={\left(\sqrt[{3}]{\upsilon }\right)}^{-1}$$
where *G*′ is the storage modulus, *G*″ is the loss modulus, *n* is the number of Maxwell elements, *G_i_* is the relaxation modulus (Pa), *G** is the complex modulus (Pa), *ω* is the angular velocity (rad·s^−1^), *R* is the gas constant (J/mol·K), *G′_p_* is plateau storage modulus (Pa), *T* is the temperature (K), *N_A_* is Avogadro number (atoms/mole), and *υ* is the number density of cross-links. 

Moreover, the sensory evaluation of cheese samples was performed by a group of 10 consumers. Acceptance, taste, texture, odor, and appearance were evaluated using a 9-point hedonic scale, with 1 = least desirable and 9 = most desirable. Cheese samples were cut into pieces 3 cm × 3 cm × 2 cm in size and placed on white plates. The pieces were tempered by holding at ambient temperature for 1 h and then presented to the consumers.

### 2.5. Shelf Life Study

The physicochemical, textural, rheological, microstructural, and sensorial properties of cheese with the optimum formula were evaluated at 3, 20, 40, and 60 days of storage time. Furthermore, the microbial properties evaluation and comparison with the control Feta-cheese was performed at 0, 20, 40, 80, and 120 days of storage time at 4 °C. The control Feta-cheese did not have SBL extract in the formulation, while it had the same rennet and starter concentration as the SBL-added cheese. 

To analyze the physicochemical properties of cheese samples, the oven drying method at 102 °C was used to assess the dry matter of Feta-cheese samples according to the IDF [19]. Fat content was measured by the Gerber method explained by Marshal [20]. Total nitrogen (TN) measurement was carried out by the Kjeldahl procedure [21]. The pH of samples was measured by a digital pH meter (Niels Bohrweg, Utrecht, The Netherlands) [20]. Ash content was determined according to the IDF [22]. Salt and acidity were determined according to AOAC [23]. Proteolysis occurring during storage was monitored by measuring the trichloroacetic acid-soluble nitrogen (TCA-SN) as described by Kuchroo and Fox [24]. Acid degree value (ADV) was determined by lipid extraction of the cheese samples and titration by alcoholic KOH using the Nuñez et al. [25] procedure. A GC–MS instrument (Shimadzu GCMS-QP-2010 plus) equipped with a Rxi-5MS fused silica column (30 m × 0.25 mm ID, 0.25 μm film thickness) was employed to analyze the chemical composition of essential oils during storage according to the Cui et al. [26] procedure. Syneresis was determined by weighting the excluded water from curd as the percentage of sample weight in the cup. 

A universal testing machine (CNS Farnell Com, Borehamwood, UK) was used to perform texture profile analyses (TPA). A 100 g cheese cup was allowed to rest at room temperature for 1 h. A 25 mm diameter perplex conical-shaped probe was employed to perform the TPA analysis. The hardness (N, maximum force of the first compression), cohesiveness (area under the second compression/ area under the first compression), adhesiveness (N, negative force area for the first compression), springiness (mm, the recovered height during the end of first compression and the start of second compression), gumminess (N, hardness × cohesiveness), and chewiness (mJ, gumminess × springiness) were determined [27,28].

Strain sweep experiments were carried out in the range of 0.01 to 1000% strain amplitude using a controlled shear rate mode at 25 °C and a constant frequency of 1 Hz. Strain sweep data were analyzed at nonlinear viscoelastic region (n-LVE). LAOS data were qualitatively evaluated through nonlinear stress waveform analysis and elastic and viscous Lissajous plots; where total stress (*τ*/*τ*_0_) was plotted versus *γ*/$\gamma_{0}$ (elastic Lissajous plots) and $\dot{\gamma }$/${\dot{\gamma }}_{0}$ (viscose Lissajous plots). The start of the nonlinear region was quantitatively evaluated by $G_{3}^{\prime \prime }$/$G_{1}^{\prime \prime }$ and $G_{3}^{\prime }/G_{1}^{\prime }$ ratios calculated from Fourier transformed analysis in time domain. Moreover, the Ewoldt et al. [29] procedure was employed to gain meaningful physical parameters. According to the Ewoldt et al. [29] method, two ratios of shear-thickening and strain-hardening from the instantaneous viscosities and elastic moduli were determined, respectively, as follows:(8)$$S=\frac{\left(G_{L}^{\prime }-G_{M}^{\prime }\right)}{G_{L}^{\prime }}$$
(9)$$T=\frac{\left(\eta_{L}^{\prime }-\eta_{M}^{\prime }\right)}{\eta_{L}^{\prime }}$$
where $\eta_{L}^{\prime }$, $G_{L}^{\prime }$, $\eta_{M}^{\prime }$ and $G_{M}^{\prime }$ point out the instantaneous viscosity at the maximum strain rate, the strain modulus at maximum strain, the instantaneous viscosity at the minimum strain rate, and the strain modulus at minimum strain, respectively. S > 0 indicates the intracycle strain stiffening. S < 0 indicates the intracycle strain softening. T > 0 represents the intracycle shear thickening and T < 0 corresponds to the intracycle shear thinning.

Dynamic frequency sweep measurements were performed in the frequency range of 0.062–62.8 rad/s within the LVE range (strain amplitude of 0.01%). All the studied rheological parameters in Section 2.4.2 were employed to evaluate the rheological properties of SBL-added cheese during a 60-day storage period.

Scanning electron microscopy (SEM) was carried out using the Karami et al. [30] method by scanning electron microscope instrument model (Leo 1450 VP, German-British). Briefly, cheeses were cut into cubes and fixed in 2.5% (*w*/*v*) glutaraldehyde solution for 3 h. Samples were refrigerated until test time. Then the samples were mounted on stubs and covered with gold–palladium. The SEM micrographs were analyzed, and the texture indices estimated by the gray-level co-occurrence matrix (GLCM) method were employed to quantitatively differentiate the texture features of the SEM images during shelf life [31]. The GLCM method computes the joint probability of two pixels to extract the second-order statistical information from the gray-level image. Before making the matrix, the d (the distance between pixels) and Ɵ (predefined particular direction) must be determined. Generally, d is determined based on the properties of food. Since there is not enough information about the appropriate distance, the 1 value was chosen. Herein, d was set as one and Ɵ was set as four directions including 0, 45, 90, and 135°. The texture indices of entropy (measures randomness of the gray level in the image), contrast (estimates local variations present in the image), and homogeneity (measures spatial similarity of the image structures) were derived from the GLCM [31]. Finally, the relationship between the instrumental texture parameters (hardness, cohesiveness, adhesiveness, springiness, gumminess, and chewiness) and the SEM texture indices were determined by linear fitting. 

Sensory properties of cheese samples were evaluated during the 60-day shelf life. A 9-point hedonic scale was employed to evaluate the sensory parameters of acceptance, taste, texture, odor, and appearance as described in Section 3.2.1.

To characterize and compare the microbial state of SBL-containing and control cheeses, initially, 25 g of each cheese type was homogenized with 225 mL sterile sodium citrate solution (2% *w*/*w*) in a Stomacher (Seward, Worthing, UK) for at least 3 min. Sequential decimal dilutions were made (10-2 and 10-7) with peptone water. LAB were determined from MRS agar (Merck, Darmstadt, Germany) incubated at 37 °C for 24 h under aerobic conditions where, cycloheximide at 0.1 g/L was added to prevent the growth of fungi. Yeast and mold were grown on YGC agar for 72 h at 27 °C. Subsequently, colony forming units (CFU) were counted in plates containing 30 to 300 colonies and results were expressed as CFU per gram of cheese. All plate counts were carried out in triplicate at the 0, 40, 80, and 120th day of storage time. To have a more accurate analysis, three differential models of J shape (Equation (10)), S shape (Equation (11)), and semicircular shape (Equation (12)), where each of them represented one type of growth pattern, were employed as follows:(10)$$\frac{d\left(P\left(t\right)\right)}{dt}=\alpha \times \left(P\left(t\right)\right)\to P\left(t\right)=P\left(t_{0}\right)\times epx\left(\alpha t\right)$$
(11)$$\frac{d\left(P\left(t\right)\right)}{dt}=\alpha \times \left(\frac{\left(P\left(t_{eq}\right)-P\left(t_{0}\right)\right)}{P\left(t_{eq}\right)}\right)\times P\left(t\right)\to P\;\left(t\right)=\frac{P\left(t_{eq}\right)}{\left(1+\left(\frac{\left(P\left(t_{eq}\right)-P\left(t_{0}\right)\right)}{P\left(t_{eq}\right)}\right)\times \exp\left(-\alpha c\right)\right)}$$
(12)$$\frac{d\left(P\left(t\right)\right)}{dt}=\alpha \times \left(P\left(t_{eq}\right)-P\left(t_{0}\right)\right)\to P\left(t\right)=\left(P\left(t_{0}\right)-P\left(t_{eq}\right)\right)\times epx\left(-\alpha c\right)+P\left(t_{eq}\right)$$
where *P*(*t*) represents the microbial population, *α* represents the rate of growth change, $P(t_{eq})$ represents the final microbial population, and $P(t_{0})$ represents the initial microbial population.

### 2.6. Statistical Analysis

Central composite design–response surface methodology (RSM) was applied to optimize the extraction process and the cheese formulation. For optimization of microwave-assisted extraction, 13 experimental runs were conducted, 5 runs as center points and 8 runs for determining the coefficient of models. For optimization of cheese formulation, 20 experimental runs were designed, 6 runs as center points and 14 runs for determining the coefficient of models. Additionally, a completely randomized design was employed to study the shelf life of cheese with the optimum formula. Paired comparisons of means were performed using the Duncan test (*p* < 0.05). Curve fitting was made by MATLAB 2010 (7.10.0, MathWorks, Natick, MA, USA), using the Levenberg–Marquardt algorithm. Agglomerative hierarchical clustering (bottom-up) was performed by MATLAB software (2010, version 7.10.0).

## 3. Results and Discussion

### 3.1. Satureja Bachtiarica Leaf Extraction

The quadratic model adequately fitted the EY, IZD, and DPPH experimental data and the parameters are shown in Table 1. There were low levels of RMSE and high levels of R^2^ and R^2^_adj_ for all the studied responses (EY, R^2^ = 0.98, R^2^_adj_ = 0.95, RMSE = 0.25; IZD, R^2^ = 0.93, R^2^_adj_ = 0.91, RMSE = 0.53; DPPH, R^2^ = 0.90, R^2^_adj_ = 0.87, RMSE = 0.96), which approved the high significance of the model. Furthermore, the coefficient of variation value was less than 3.58% for all the models which indicated acceptable reproducibility. The linear effect of time and power were significant on EY, IZD, and DPPH parameters (*p* < 0.05). Furthermore, the interactive effect of time–power, the quadratic effect of time, and the quadratic effect of temperature on the DPPH parameter, as well as the quadratic effect of time on IZD were significant (*p* < 0.05). Zekovic et al. [8] reported that, in microwave-assisted extraction of *Satureja montana* L. extract, the linear term of extraction time was the most dominant factor in the total flavonoids content of the extract among the three factors of extraction time, ethanol concentration, and irradiation power. The response surface plot of time–power interactive effect on DPPH showed that at high power value (800 W) the DPPH increased with time of microwave exposure (Figure 1). 

EY was the most affected variable for each 100 W increase in power or for each 1 min increase in microwave time, compared with the DPPH and IZD parameters (data not shown). The best microwave power and time were determined as 800 W and 8 min, respectively. The desirability value was obtained 98% which supported adequate accuracy of optimization. During 8 min microwave at 800 W power, the EY, IZD, and DPPH were 16%, 73.56 mm, and 24.2%, respectively. Under these optimal conditions, the experimental values were in compliance with the predicted values by analysis of variance at the level of 0.05. 

### 3.2. Cheese Formulation

#### 3.2.1. Rheological and Textural Properties

One of the most important determinants of Feta-cheese quality is the rheological properties. The dynamic oscillatory frequency sweep test is a useful and rapid technique for assessing the rheological features of cheese [32]. Using the frequency sweep technique, the effects of SBL extract, starter, and rennet concentrations were studied on the rheological properties of Feta-cheese and, finally, 10 rheological parameters were generated, i.e., *k*′, *k*″, *n*′, *n*″, *k*′/*k*″, *z*, *A*, *ξ*, $\lambda_{rel}$, and *η*_s_*. In the studied frequency range, elastic modulus (*G*′) dominated viscose modulus (*G*″) and showed weak frequency dependency without a crossover point which suggested a weak gel-like behavior of all cheese samples [33]. It was similar to the results reported by Omrani Khiabanian et al. [12] and Hammam et al. [11] for UF white cheese. To simplify the discussion, 10 generated rheological parameters were clustered in four groups with more than 83% similarity indices (SI) in each category by the agglomerative hierarchical clustering (bottom-up) method (Table 2). These groups were based on properties related to the strength of linkages (*A*), the number of linkages (*z*), the timescale of junction zone ($\lambda_{rel}$), and the distance of linkages (*ξ*) [33]. 

Model fitting of the results showed that all the rheological parameters (*z*, *A*, *ξ*, and $\lambda_{rel}$) followed the quadratic model with R^2^ > 0.88, R^2^_adj_ > 0.83, RMSE < 1.52, CV < 7.89, and nonsignificant lack of fit. The RSM model parameters (interactive, quadratic, and single effects) of different rheological parameters are presented in Table 3.

ANOVA results for the strength of linkage parameter (*A*) showed that the *A* parameter was dependent on the linear effect of X_1_ (extract concentration), X_2_ (starter concentration), and X_3_ (rennet concentration); the interaction effect of X_1_X_2_, X_1_X_3_, and X_2_X_3_, as well as the quadratic coefficient of X_1_^2^ and X_3_^2^ variables. At low starter concentration, *A* parameter increased with an increase in rennet concentration (Figure 2a). Rennet addition to the milk results in a high aggregation rate of casein and consequently a coarse network formation, which is responsible for the firm structure of Feta-cheese. In the early stage of Feta-cheese production, this strengthening effect is dominated by the softening effect of the proteolysis [30]. On the other hand, at high starter concentration, *A* parameter decreased with an increase in rennet concentration (Figure 2a). This behavior could be explained by the increase in cheese proteolysis due to the excess amount of rennet when both starter and rennet concentrations were high which resulted in the decrease in the Feta-cheese strength. Hesari et al. [34] reported that both rennet and starter are responsible for the proteolysis of UF white cheese. Furthermore, at the constant rennet or starter concentration, the *A* parameter decreased with an increase in SBL extract concentration. The reason for this phenomenon could be the native proteolytic enzymes of the herb extract which decreased the cheese strength by increasing the proteolysis. Similarly, Solhi et al. [35] reported an increase in processed cheese proteolysis with an increase in asparagus powder concentration and related this phenomenon to proteolytic enzymes of asparagus powder. Another reason could be that the extract content interfered on the rennet action and retarded the pH reduction, as was shown by the higher pH of samples with the greater SBL extract concentration (data not shown). pH has a dominant role in the viscoelastic properties of Feta-cheese. The lower pH of Feta-cheese results in the dominant elastic character and hardening of cheese [30]. At low rennet concentration or high SBL extract concentrations, the *A* parameter increased with the increase in starter concentration (Figure 2a,b). This behavior could be related to the pH decreasing effect of starter which resulted in the greater strength of the cheese network.

ANOVA results for the network extensity parameter (*z*) showed that *z* parameter was dependent on the linear effects of all variables, the interactions of X_1_X_2_ and X_1_X_3_ and the quadratic effect of X_3_^2^; whereas it was independent of the interactions of X_2_X_3_ and the quadratic effects of X_2_^2^ and X_1_^2^ (Table 3). Increasing the SBL extract at high starter concentration abruptly decreased *z* parameter, while this change was not significant at low starter concentration. High *z* value of a cheese sample represented greater interaction between cheese components, which would induce a greater density of subunit groups [36]. Furthermore, at low SBL extract concentration, the *z* parameter increased with an increase in starter concentration. This behavior could be explained by a decrease in pH of cheese by starter, which was not retarded by the SBL extract, since the SBL extract concentration was low. 

ANOVA results for the relaxation time ($\lambda_{rel}$) showed that the $\lambda_{rel}$ was dependent on the linear effects of all the variables, the interaction effect of X_1_X_3_, and the quadratic effect of X_3_^2^ (Table 3). At all the starter concentrations, the $\lambda_{rel}$ value decreased when increasing the SBL extract concentration in the cheese (Table 3). This behavior suggested that the increase of SBL extract in cheese samples decreased the mean lifetime of the junction zones of entanglements within cheese structural components and could be related to dissociation of intermolecular aggregates [18]. 

ANOVA results for *ξ* parameter showed that the *ξ* parameter was dependent on the linear effect and the interaction effect of all variables and the quadratic effect of X_1_^2^ (Table 3). At the constant rennet or starter concentration, *ξ* increased with an increase in SBL extract concentration. *ξ* represents the distance between sequential crosslinking points in a material structure [36].

#### 3.2.2. Sensorial Property

Model fitting of the sensory properties showed that the texture score followed the quadratic model; while odor, acceptance, and taste followed a linear model with R^2^ ≥ 0.86, R^2^_adj_ ≥ 0.83, RMSE ≤ 1.62, CV ≤ 4.5 and nonsignificant lack of fit. It is worth mentioning that appearance did not follow any model, significantly. The taste and odor were dependent only on the linear effects of X_1_ and X_2_ (Table 3). The scores of these responses decreased with an increase in SBL extract concentration. The texture score was dependent on the linear effects of X_1_ and X_3_, the interaction effect of X_1_X_2_, X_1_X_3_, and X_2_X_3_ and the quadratic effect of X_1_^2^. Furthermore, appearance was only dependent on the linear effect of X_1_. The effect of each 0.1 g/100 kg SBL extract concentration on odor, acceptance, taste, appearance, and texture was not significant in the range of 0.3 to 0.6 g/kg; whereas further increase in extract concentration diminished the scores of these parameters (data not shown). Herb extracts often have a strong flavor even when used in small concentrations, which can also result in rejection of the flavored cheese, turning it into a limiting factor in the use of plants and their constituents [37]. Therefore, although using SBL extract at 0.6 g/100 kg concentration did not have a bad sensory impact, higher-level usage is not recommended. Fadavi and Beglaryan [1] reported that peppermint extract addition showed a negative effect on the sensory score of UF Feta-type cheese. Each 0.1 g/100 kg increase in rennet concentration did not significantly change the odor, acceptance, and taste scores; whereas it increased the texture score when it was added in the range of 1.5–2 g/100 kg and decreased the texture score when it was added in the range of 2–2.5 g/100 kg (data not shown). Fadavi and Beglaryan [1] found a similar relationship between the taste of Feta-cheese and rennet concentration. Although, starter concentration did not significantly affect the texture and appearance scores, each 0.1 g/100 kg increase in starter concentration in the range of 1.5–2.9 g/100 kg resulted in an increase in odor, acceptance, and taste. It is known that lactic acid starter by means of carbohydrate metabolism, lipolysis, and proteolysis produces the typical flavor of cheese [10]. 

#### 3.2.3. Optimization of Cheese Formulation

Among the evaluated parameters in our experiments, some responses were selected to optimize the cheese formulation. The criteria were maximum SBL extract concentration usage and maximum sensory scores for odor, acceptance, taste, appearance, and texture. Maximum desirability was predicted to be 91% at 1.95 g/100 kg rennet, 2.2 g/100 kg starter, and 0.45 g/100 kg SBL extract concentrations. At the optimized conditions, the odor, acceptance, taste, texture, appearance, $\lambda_{rel}$, *ξ*, *z*, and *A* were 8.05, 8.13, 8.07, 8.15, 8.15, 258.70 s, 5.41 nm, 3.03, and 34,036.12 Pa·s^1/z^, respectively. The actual data and the results of the selected model did not display any significant difference at the level of 0.05.

### 3.3. Storage study

#### 3.3.1. Physicochemical Properties

Table 4 shows the effect of storage time on pH, dry matter, fat, ash, salt, syneresis, trichloroacetic acid-soluble nitrogen/total nitrogen ratio (TCA-SN/TN), and the acidity number (ADV) of SBL-added UF Feta-type cheese at the optimal formulation. The chemical composition of the cheese sample met the specifications for the first quality Feta-cheese as described by Iranian standards for UF Feta-type cheese indicating that the addition of SBL extract did not adversely affect the UF Feta-type cheese composition. Any syneresis was not observed during a 60-day period of storage. Similarly, Akbarian Moghari et al. [38] reported insignificant change in syneresis of fortified Feta-cheese by probiotic bacteria between the 1st and 60th day. During storage time, dry matter, salt, and ash levels of the samples remained relatively constant (*p* > 0.05). Likewise, Karami et al. [30] did not report any changes in dry matter and salt level of UF Feta-type cheese during 60-day shelf life. pH level decreased from the 3rd day (4.89) to the 40th day (4.65); whereas, at the 60th day, it increased to the initial level (4.90). Starter activity during ripening results in curd acidification by conversion of lactose to lactic acid [11]; while the released amino acids during cheese ripening raise the pH value to some extent [30]. Buffering in cheese could also be related to the presence of proteins and inorganic constituents such as weak acids, bases, and metal ion complexes [39]. Solhi et al. [35] did not observe any change in the pH of processed cheese containing tomato powder before 40th day of storage but there was an increase in the pH after that time. Shan et al. [40] reported that the increase in pH during storage is inhibited by herbal extracts due to their high quantity of phenolic composition. 

Titratable acidity of the samples increased from 0.89% at 3 days of storage to 1.02% at 60 days of storage. Similar to our results, Shahab Lavasani et al. [41] reported an increase in acidity due to the lactose conversion into lactic acid by the starter culture. The nitrogen soluble in 12% trichloroacetic acid (TCA-SN)/ Total nitrogen (TN) ratio is an indicator of proteolytic reaction. The TCA-SN/TN ratio did not change within 20 days of storage, whereas it increased from 20 days to 60 days of storage. This behavior suggested that extending the storage time led to an increase in protein degradation in Feta-cheeses. It seems that proteolysis was more prominent during the last stages of storage. Proteolytic reaction can produce proteose peptones, peptides, whey proteins, and free amino acids. Rennet, plasmin, and microbial peptidases could produce these nitrogen compounds [41]. By investigating the TCA-SN level of asparagus-powder-added processed cheese, Solhi et al. [35] reported an increase in TCA-SN level during a 90-day storage and related this phenomenon to thermoduric microbiota or native vegetable proteolytic enzymes of the asparagus powder which was added to cheese. Compared to Karami, et al. [30], for traditional Feta-cheese, the intensity of proteolysis for UF Feta-type cheese obtained in this study was higher. Addition of herb to cheese enhances the proteolysis which accelerates the ripening of herby cheese [42]. Tarakci et al. [43] found black cumin (*Nigella sativa*) addition to Tulum cheese increased the level of water-soluble nitrogen. A decrease in proteolysis of UF Feta-type cheese results in textural and sensorial defects [30]. 

The total fat content decreased, and the lipolysis index (mEq/g) increased within 20-day storage, while they did not change after that time, significantly. Since milk lipase is inactivated by pasteurization (76 °C for 15 s) and during the ultrafiltration (50 °C for 30 min), it seems that LAB (starter) is the main lipolytic agent maintained during the ripening stage of UF Feta-type cheese. Lipolysis induces free fatty acids (FFA), which contribute to cheese flavor and could be precursors of flavor compounds such as alcohols, methyl ketones, and lactones [44].

Although the overall level of free fatty acid, as evidenced by ADV, increased after 60 days, the relative amount of each fatty acid in fatty acid profile was different (Table 5). The long-chain free fatty acids (palmitic acid, stearic acid, oleic acid, linoleic acid, arachidic, and linolenic acid) were the predominant free fatty acids in Feta-cheese followed by medium-chain FFA included capric acid, lauric acid, and myristic acid. Palmitic and oleic acid dominated among the saturated and unsaturated long-chain FFAs of Feta-cheese, respectively. Similarly, Temiz et al. [44] found palmitic and oleic acids as the most abundant FFAs in herby brined cheese types. Although, the other volatile (C4:0 to C8:0) fatty acids were far lower than long-chain and medium-chain FFAs, the former FFAs intrinsically contribute to cheese flavor, since they have the lowest perception thresholds [45]. The butyric acid content (1.8 g/100 g at the 3rd day) of Feta-cheese in the present study was higher than 0.85 g/100 g which was found by Katsiari et al. [46] for industrially made Feta-cheese. The fatty acid composition of SBL-extract-added Feta-cheese significantly changed during 60 days of storage at 4 °C (Table 5). C4:0 to C14:0 increased up to 40 days, whereas after that time, they decreased. Cheese can suffer hydrolytic and oxidative rancidity during shelf life, resulting in the release of volatile fatty acids (C4:0–C10:0) which are responsible for the bad odor and taste of cheese [47]. The disappearance of some FFAs during cheese ripening suggested they were catabolized [44]. At the early stage of ripening, total small- and medium-chain free fatty acids of cheese were 2.8 and 12.39% of fat, respectively; however, they increased to 4.14 and 15.67%, respectively, at the 60th day. On the other hand, the long-chain fatty acids decreased from 80 to 75.25% during 60 days of storage. Up to the 40th day, the relative percentage of saturated fatty acids such as lauric and myristic acids increased; whereas unsaturated fatty acids including C18:1 and C18:2 decreased. During lipid oxidation, the double bond oxidation in unsaturated fatty acids resulted in an increase of saturated to unsaturated fatty acids proportions [47]. Similarly, Khan et al. [48] observed an increase in short-chain fatty acids and a decrease in long-chain unsaturated fatty acids during the storage of Gouda cheese. Bin et al. [40] found that herb extracts (cinnamon stick, oregano, clove, pomegranate peel, and grape seed) increased the stability of cheese against lipid oxidation; while the extracts exhibited lower inhibitory effects when added to cheese than in media systems. 

#### 3.3.2. Rheological Properties

##### Strain Sweep Test in Nonlinear Viscoelastic Region (LAOS)

In n-LVE region, material behavior depends on the magnitude and the rate of stress/strain [33]. The elastic and viscous Lissajous-Bowditch plots at LVE region and n-LVE region (Figure 3) and waveform plots (Figure 4) are illustrated. 

During storage, the shape of the elastic and viscous Lissajous plots of cheese samples was perfectly elliptical at 0.01% strain (in LVE), demonstrating ideal viscoelastic behavior (data not shown). With the increase in strain, the encompassed area of the elastic Lissajous plots increased and those of viscous Lissajous plots decreased. The elastic Lissajous plots stretched horizontally and flattened vertically tending to become a squarelike shape, representing the shift from elastic to viscous-dominated behavior (Figure 3). Furthermore, the rectangular shape of viscous Lissajous plots indicates a shear-thinning behavior of cheese samples during 60 days of storage. LAOS flow was also qualitatively investigated via nonlinear shear stress waveforms. We employed this procedure to distinguish changes in cheese structural topology during storage [49]. Nonlinear shear stress waveforms of cheese samples at the 3, 20, 40, and 60th days of storage at 1000% strain are shown in Figure 4. According to this figure, cheese at the 3rd day showed a triangular shape, at the 20th day the triangle shape was more obvious, at the 40th day it showed a forward-tilted shape, and finally, at the 60th day it showed a rectangular waveform shape. Alghooneh et al. [36] found high correlation between network extensity parameter (*z*) and stress waveforms. Similarly, the *z* parameter slightly increased from the 3rd day to the 20th day storage, while it decreased with further increase in storage (Part “Frequency Sweep Test”), which indicated a decrease in cheese network extension at the end of storage time correlated with the shape of waveforms. These results suggested that shear stress waveforms data could help us differentiate nonlinear behavior of cheese during storage.

To quantitatively evaluate the nonlinear viscoelastic responses of cheese samples during storage, the stress response to a strain input was written as a Fourier series and the ratio of the third harmonic viscoelastic modulus to the first harmonic viscoelastic modulus, $G_{3}^{\prime }/G_{1}^{\prime }$ and $G_{3}^{\prime \prime }/G_{1}^{\prime \prime }$, were determined (Table 6). The $G_{3}^{\prime }/G_{1}^{\prime }$ or $G_{3}^{\prime \prime }/G_{1}^{\prime \prime }$ higher than 0.01 reflects the start of nonlinear viscoelastic behavior [50]. The start of nonlinear behavior of SBL-added cheese at 3rd day was at 1% strain, at the 20th day was at 0.1% strain and at 40th day and 60th day was at 10% strain (Table 6). This behavior could be explained by the longest timescale of interaction between cheese components at the 20th day of storage and indicated that the required time for a new structural unit to replace those disrupted by small deformations in the strain sweep experiment was the longest at this stage of storage [51]. The increase in critical strain, at which nonlinear behavior of SBL-added cheese started, with storage time could be explained by the proteolytic action of starter microorganisms which affected the rheological properties of cheese by the breakdown of caseins over time. Proteolysis results in the reorganization and weakening of the casein matrix [30].

Strain-hardening (S) and shear-thickening (T) ratios were determined from the elastic moduli and instantaneous viscosities, respectively (Table 6). Beyond the linear region, cheese samples showed intracycle strain stiffening (S > 0) and intracycle shear thinning behavior (T < 0) at all the storage dates (Table 6). The strain-stiffening and shear-thinning behaviors of a material, simultaneously, suggested that the breakdown behavior of the network structure involves stretching of the structuring unit (strain-stiffening) followed by breaking the structure-holding interactions, resulted in shear-thinning behavior [52].

##### Frequency Sweep Test

Power-law equation (Equations (1) and (2)) satisfactorily fitted the results of *G*′ vs. *ω* and *G*″ vs. *ω* (R^2^ = 0.91–0.96 and RMSE = 0.24–0.68). Throughout the tested frequency range, the elastic behavior of all the stored cheese samples dominated over the viscous behavior (data not shown), which suggested the character of a solidlike cheese in the linear viscoelastic range during shelf life [18]. Similarly, Farbod et al. [9] and Meza et al. [39] reported greater elastic modulus than viscous modulus during 60-day storage of Feta-cheese. Wium et al. [53] claimed that the dominant elastic behavior of Feta-cheese is probably due to its low pH value. Up to the 20th day, *A*, *z* increased significantly, whereas *ξ* parameter decreased (Table 7). After that time, *A*, *z* and $\lambda_{rel}$ significantly decreased with time, whereas *ξ* showed opposite behavior. This behavior could be explained by an increase in proteolysis during storage, which resulted in a decrease in cheese strength, network extension, and the timescale of interaction in cheese structure, while the distance between the sequential linkages increased. The progress in proteolysis contributes to the softening of cheese as most casein breakdown products are water-soluble and they may not participate in the framework provided by the protein matrix [39]. The highest relaxation time was in accordance with the lowest strain at which the n-LVE region started at the 20th day storage time (Part “Strain Sweep Test in Non-Linear Viscoelastic Region (LAOS”). By studying the storage and loss moduli of Feta-cheese, Karami et al. [30] found that both moduli significantly increased as the ripening continued, and they related this behavior to a decrease in pH during ripening, which was not the case in our study. 

#### 3.3.3. Textural Properties

Cheese texture is related to the arrangement of various chemical components such as fat and protein within distinct micro- and macrostructure levels [54]. The textural properties of SBL cheese during shelf life are presented in Table 7. Springiness (mm), hardness, gumminess, chewiness, and cohesiveness values increased from the 3rd day to the 20th day; whereas the parameters decreased with further increase in shelf life. Similarly, Omrani Khiabanian et al. [12] reported an increase in springiness (mm), hardness, gumminess, chewiness, and cohesiveness of Feta-cheese with increase in shelf life from 1 to 30 days. The cheese hardness is represented by the maximum force exerted on its cube to achieve 70% stress compaction with respect to its original size [9]. A decrease in hardness, gumminess, adhesiveness, chewiness, and cohesiveness at the end of storage time could be due to an increase in proteolysis of SBL-containing Feta-cheese during shelf life. Farbod et al. [9] found that at the 2nd month of Feta-cheese storage, an increase in proteolysis by proteinase and peptidase activities of lactic acid bacteria happens, which results in softer cheese. This observation is in agreement with the results of Katsiari et al. [55] for Feta-cheese. This observation confirms the importance of proteins in cheese texture that was previously published by others [39]. Mohamed et al. [56] found that the hardness and cohesiveness of Moringa oleifera-extract-added cheese increased during storage; whereas the springiness, gumminess, and chewiness decreased with increase in storage time. They explained this behavior by a decrease in pH value. Different results in our study could be explained by the fact that the change in pH of cheese was not significant between the 3rd day and the 60th day. 

#### 3.3.4. Microstructural Properties

The quality attributes of cheese are linked to its microstructure. Investigation of Feta-cheese microstructure could help us better interpret the cheese’s rheological properties. The SEM images allow us to directly detect the changes of the material matrix in micro-scale which contributes substantially to our understanding of structure–function relationships [31]. At the 3rd storage day, fat globules were clearly observed in the cheese matrix (Figure 5a). SEM of Feta-cheese displayed the disruption of fat globules during the ripening period. At the 60th day of storage, SEM images showed a homogenous and compact texture without any fat globule (Figure 5d), which could be related to the progressive lipolysis during storage (Section 3.3.1). A similar trend was observed by Karami et al. [30] and Omrani Khiabanian et al. [12] for Feta-cheese. Studies on cheeses with different fat levels showed that an increase in entrapped fat globules in the protein matrix decreases the hardness value [57]. Fat globules disrupt the uniformity of the gel structure and increase the number of weak points in the cheese structure [30]. This image could confirm the lower springiness (mm), hardness, gumminess, chewiness, and cohesiveness values of cheese sample at the 3rd day storage than at the 20th day storage (Section 3.3.3). On the other hand, the progressive decrease in these parameters from the 20th day to the 60th day storage was related to the increase in proteolysis during storage which was not interpretable by the extent of fat globules in SEM micrographs. Therefore, we need a quantitative analysis of SEM micrographs. To quantitatively describe the texture shown in the SEM images and detect the relationship between the cheese mechanical properties and SEM images, indices estimated by the gray-level co-occurrence matrix (GLCM) method were employed. The texture analysis of the SEM images showed that the highest values of entropy and contrast happened at the 20th day of storage (11.90 ± 0.89 and 52.06 ± 3.87, respectively), while the highest homogeneity level was obtained at the 60th day of storage (8.41 ± 0.70) (data not shown). The more monotonous of the image gray levels results in the lower the entropy and contrast, and the higher the homogeneity [31]. The indices estimated from cheese samples were used to determine their relationship with mechanical properties by linear fitting. Image texture features calculated from GLCM (homogeneity, contrast, and entropy) at all storage dates had high correlations with instrumental mechanical characteristics, i.e., chewiness, hardness, cohesiveness, springiness, gumminess, and adhesiveness (R^2^ = 0.93–0.97, RMSE = 0.23–1.23) (Table 8). These results suggested that SEM image texture indices can be useful tools to determine the mechanical parameters behavior of Feta-cheese during shelf life. Pieniazek and Messina [58] found a positive correlation between the tenderness of semitendinosus and gluteus medius bovine muscles and image features of energy and homogeneity. Evaluating the mechanical damage on wheat starch granules by SEM, Barrera et al. [59] reported that the surface of the damaged granules presented higher entropy and lower homogeneity values than undamaged starch.

#### 3.3.5. Sensorial Properties

Storage did not a1w2q1ffect the appearance score during a 60-day period (Table 7) which could be due to syneresis phenomenon did not happening during storage (Section 3.3.1). Moreover, the SBL extract was colorless, so it did not result in off-color during storage. The texture score was at the highest value at the 20th day, whereas it decreased with further increase in storage time up to the 60th day. This result could be explained by the increase in lipolysis and proteolysis after the 20th day (Section 3.3.1). Acceptance, odor, and taste were at the highest value at 40 days of storage. The increase in cheese odor up to the 40th day could be related to an increase in lipolysis, as was shown by the ADV number, an increase in small-chain free fatty acids at the early stage of storage (Section 3.3.1); and the high metabolic activities of LAB that may produce flavor compounds [10]; whereas, at the end of storage, they diminish due to volatility (Section 3.3.1). Although the scores of taste and odor decreased from the 40th day to the 60th day, they had relatively high scores and panelists were generally satisfied. Comparing different herb extracts, Tayel et al. [60] found that ethanolic cinnamon extract (70%) was the most desirable for taste and overall quality enhancement of flavored cheeses, while lemon grass improved the cheese odor. Akbarian Moghari et al. [38] found that the flavor intensity, odor, texture, and appearance of the inulin-added Feta-cheese did not change during 60 days of storage.

#### 3.3.6. Microbiological Properties

LAB viability is one of the most important measures of the quality of functional foods [38]. The possible effects of SBL extract on starter and nonstarter organisms in cheese need to be considered. Therefore, the microbial properties of the SBL-containing cheese and industrial cheese samples were compared over 120 days of storage (Figure 6).

Throughout the storage period, the probiotic bacteria (LAB) were able to maintain their viability in both the UF Feta-type cheese and viable counts were above the recommended threshold level (>6 log CFU/g) for probiotic products [38]. The LAB populations with time followed the S shape model (R^2^ = 0.96, R^2^_adj_ = 0.93, RMSE = 0.74). Although, there was not any significant difference in LAB counts of control cheese compared with SBL-added cheese up to the 40th day of storage; afterward, the LAB counts of SBL-containing cheese and control cheese increased to 8.90 and 8.65 log per gram at the 80th day, respectively (Table 9). The change in LAB counts was not significant between the 80th day and the 120th day for both cheese samples. This behavior suggested that the supportive effect of SBL natural extract on LAB viability mainly occurred between the 40th and 80th days of storage. Hamdy et al. [10] found an increase in LAB count of Feta-cheese up to 15 days storage time and a decrease in LAB count after that. They explained this behavior as due to high acidity levels during the storage time which limits the growth of microorganisms. Mohamed et al. [56] studied the effect of *Moringa oleifera* leaf extract on *Lactobacillus plantarum* population in a cream cheese and reported that with increase in extract, the *Lactobacillus plantarum* count increased. Licón et al. [13] reported that essential oil of *Melissa officinalis* was not appropriate as antimicrobial agent in pressed ewes’ cheese since it showed an inhibitory effect against LAB starter cultures. Furthermore, they found *Thymus vulgaris* essential oil inhibited the growth of *Penicillium verrucosum* without affecting the natural LAB present in the cheese.

The mold and yeast populations with time followed the semicircular shape model (R^2^ = 0.92, R^2^_adj_ = 0.89, RMSE = 1.25). SBL-containing cheese showed lower initial (*P*_0_) and final (*P*_∞_) mold and yeast populations than the control one, while the microbial growth rate was almost half of the control cheese. As discussed earlier, SBL extract is perceived to have antifungal activity [61]. The total yeast and mold count of control cheeses was more than 102 CFU/g at the end of 120 days of storage; whereas in the SBL-added cheese, it was between 101 and 102 CFU/g. Belewu et al. [62] found that the ginger extract was a highly active agent to reduce the microbial load and prolong the shelf life of West African soft cheese for 15 days. In another study, Tarakci and Temiz [63] found that Turkish Otlu-herby cheese flavored by species such as *Allium* sp., *Thymus* sp., *Mentha* sp. *Ferula* sp., and *Pranges* sp. enhanced cheese shelf stability. Bin et al. [54] investigated the antibacterial efficiency of five herb extracts (cinnamon stick, oregano, clove, pomegranate peel, and grape seed) against *Listeria monocytogenes*, *Staphylococcus aureus*, and *Salmonella enteric* and found that all five extracts inhibited the growth of the three food borne pathogens in cheddar cheese throughout the 9 days of storage. Although the yeast and molds were not eliminated completely, SBL extract was able to effectively decrease their growth rate in cheese which is particularly useful in terms of food safety for short-term storage of products. These results showed that SBL extract could control the putrefying microbes and can improve the growth of the favored microbes at the same time.

## 4. Conclusions

In this study, the main objective was to develop a nutraceutical UF Feta-type cheese containing *Satureja bachtiarica* leaf extract, with high acceptable rheological, textural, microbial, physicochemical, and sensorial properties. RSM and the conventional graphic and desirability functions methods were effective to determine the optimum zone within the experimental region of SBL extraction and cheese formulation. Microwave-assisted extraction at optimum conditions (8 min and 800 W power), exhibited the highest EY, DPPH, and IZD values, making it a promising method for obtaining valuable extract from *Satureja bachtiarica* leaves. The achieved extract was employed to determine the best cheese formula with the highest sensorial scores. The optimized cheese had 1.95 g/100 kg rennet, 2.2 g/100 kg starter, and 0.45 g/100 kg SBL extract concentrations in the formulation. During a 60-day storage period, the optimized formulated cheese did not show a significant change in syneresis, salt, ash, and dry matter. Although the pH value changed with time, there was not any significant difference in pH of three-day-old cheese and sixty-day-old cheese samples. Furthermore, the TCA-SN/TN, acid number, and the FFA profiles obtained by GC–MS technique showed an increase in proteolysis and lipolysis during storage. From the 3rd day to the 40th day of storage, the relative percentage of saturated fatty acids such as lauric and myristic acids increased; whilst unsaturated fatty acids decreased. Rheological and textural characterization of SBL-added cheese during a 60-day storage time showed the lowest network strength, timescale of interaction, network extension, hardness, gumminess, adhesiveness, chewiness, and cohesiveness and the greatest distance between the sequential linkages at the end of the storage period. This behavior was explained by an increase in proteolysis during storage. A correlation was observed between the SEM texture indices and mechanical parameters (hardness, cohesiveness, gumminess, chewiness, and adhesiveness). Results showed that the texture analysis on SEM images is a feasible method to estimate the behavior of mechanical properties of Feta-cheese during storage.

Acceptance, odor, and taste were at the highest value at 40 days of storage. The optimized cheese was compared with the control one regarding their microbiological characteristics using three novel differential models. LAB population followed the S shape model; while the mold and yeast population followed the semicircular shape model. The final populations of mold and yeast and their growth rate in the SBL-extract-added cheese was significantly lower than the control one. On the other hand, SBL extract increased the LAB counts between the 80th day and the 120th day more than the control one. In conclusion, SBL extract could be introduced as a natural antimicrobial and antioxidant additive, since SBL-extract-added cheese not only showed acceptable rheological, textural, sensorial, and physicochemical properties, but SBL extract also improved the microbiological properties of it.

## Figures and Tables

**Figure 1 foods-11-01728-f001:**
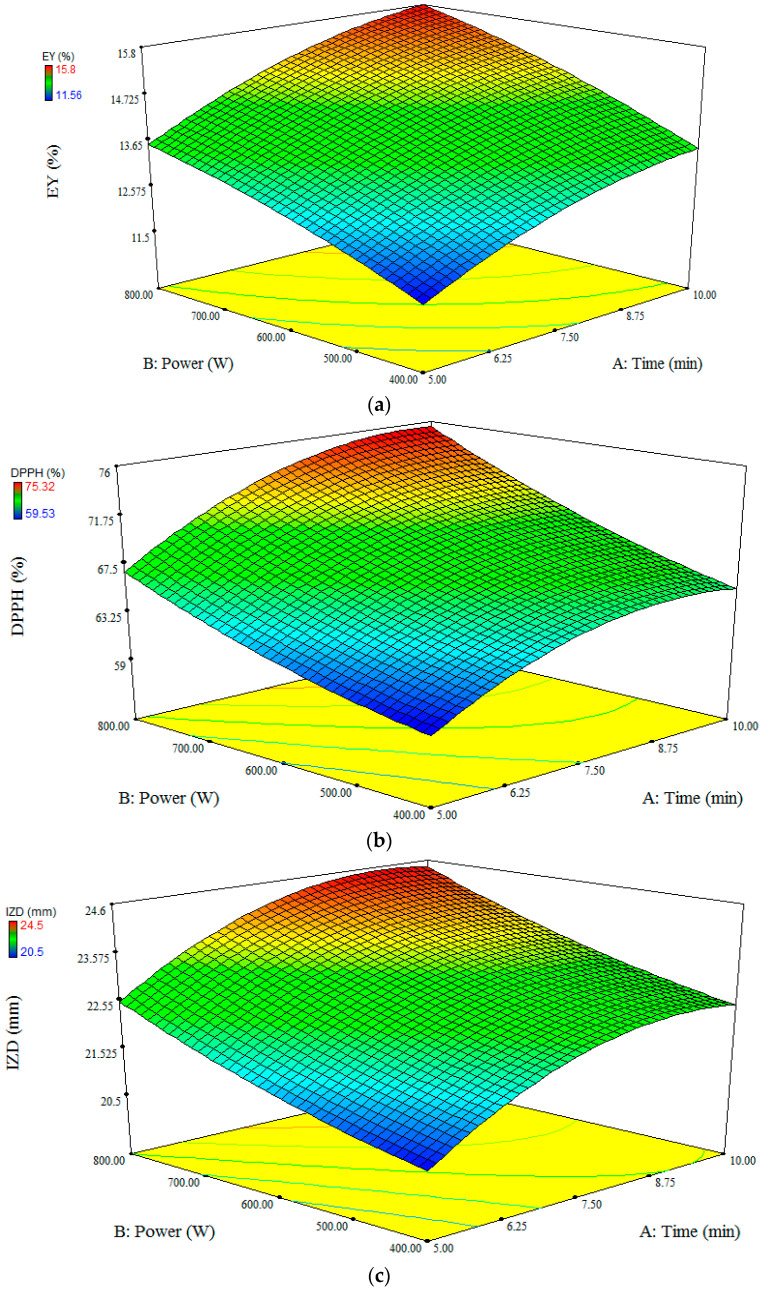
The surface response plot of the interactions of power and time on extraction yield (EY, (**a**)), inhibition zone diameter (IZD, (**b**)), and antioxidant activities (DPPH, (**c**)) of SBL extract obtained by microwave-assisted extraction method.

**Figure 2 foods-11-01728-f002:**
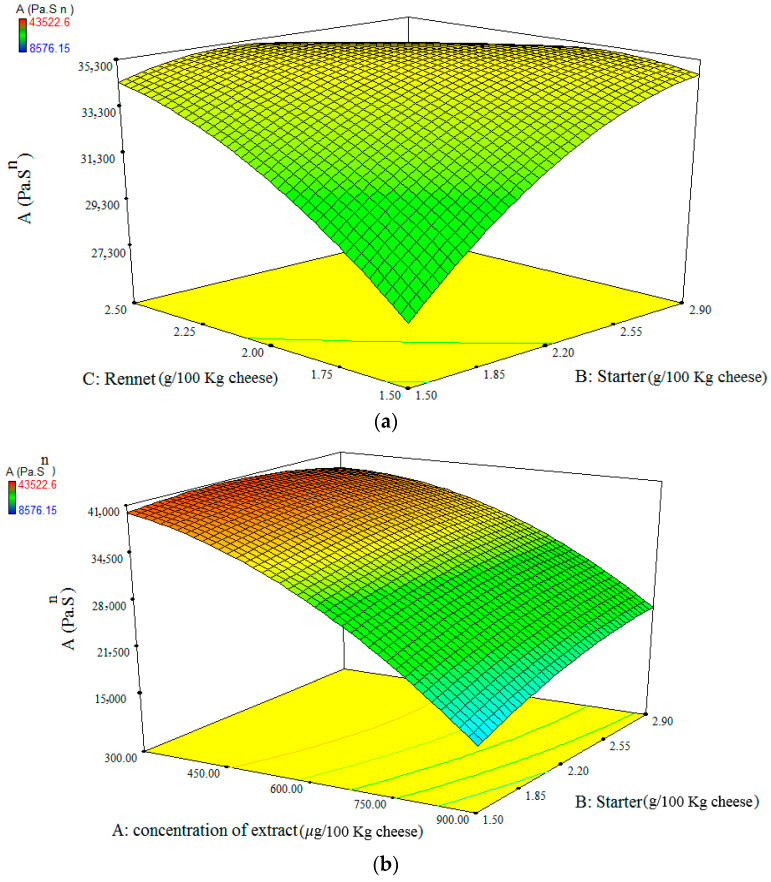
The surface response plot of the interactions of rennet and starter concentrations (**a**) and extract and starter concentrations (**b**) on the strength of the linkage parameter (*A*, Pa·s^1/z^) of SBL-extract-added cheese.

**Figure 3 foods-11-01728-f003:**
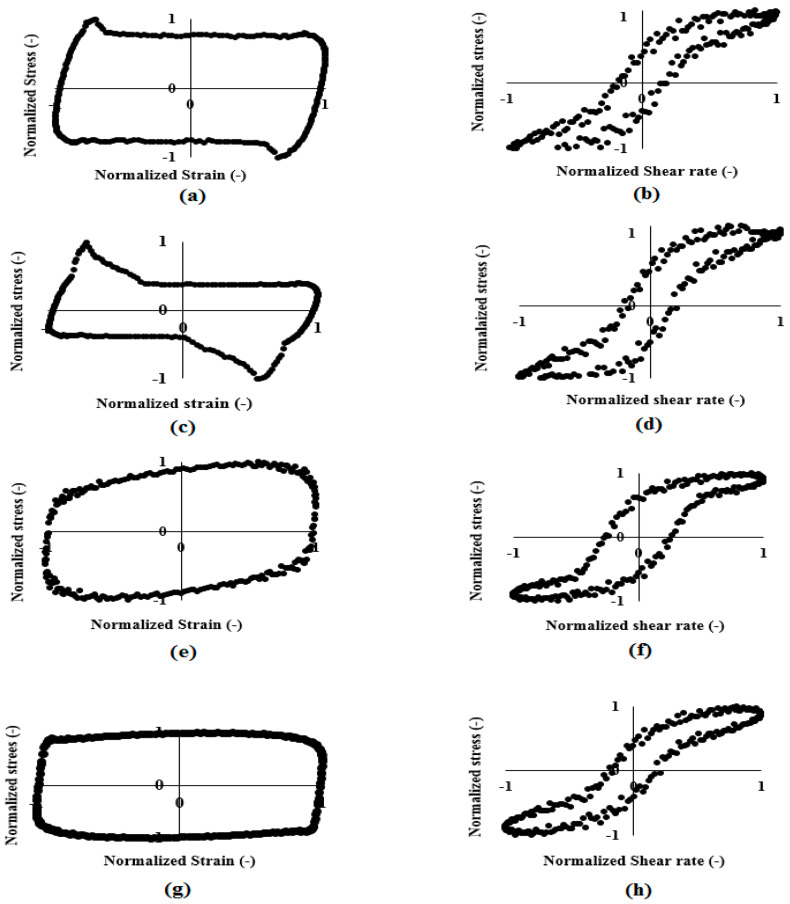
Elastic Lissajous plots of the SBL-extract-added Feta-cheese samples at 3rd day (**a**), 20th day (**b**), 40th day (**c**), and 60th day (**d**) and the viscous Lissajous plots at 3rd day (**e**), 20th day (**f**), 40th day (**g**), and 60th day (**h**) in n-LVE region (1 Hz and 1000% strain).

**Figure 4 foods-11-01728-f004:**
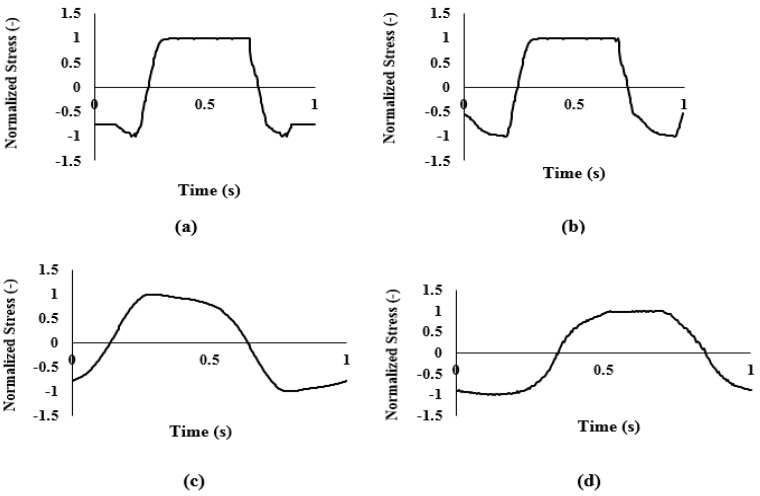
Normalized stress signal as a function of time for the SBL-extract-added Feta-cheese samples at 3rd day (**a**), 20th day (**b**), 40th day (**c**), and 60th day (**d**) in n-LVE region (1 Hz and 1000% strain).

**Figure 5 foods-11-01728-f005:**
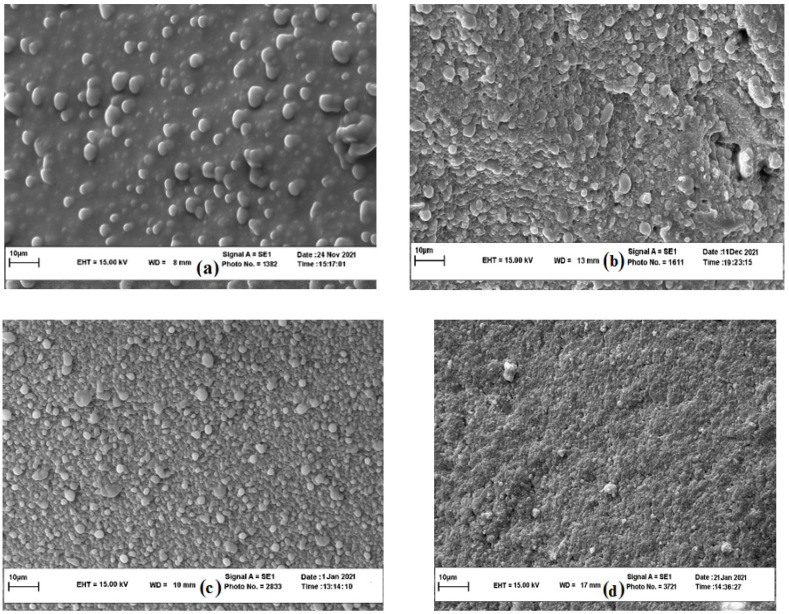
SEM images of UF Feta-type cheese at 3 (**a**), 20 (**b**), 40 (**c**), and 60 (**d**) days of storage.

**Figure 6 foods-11-01728-f006:**
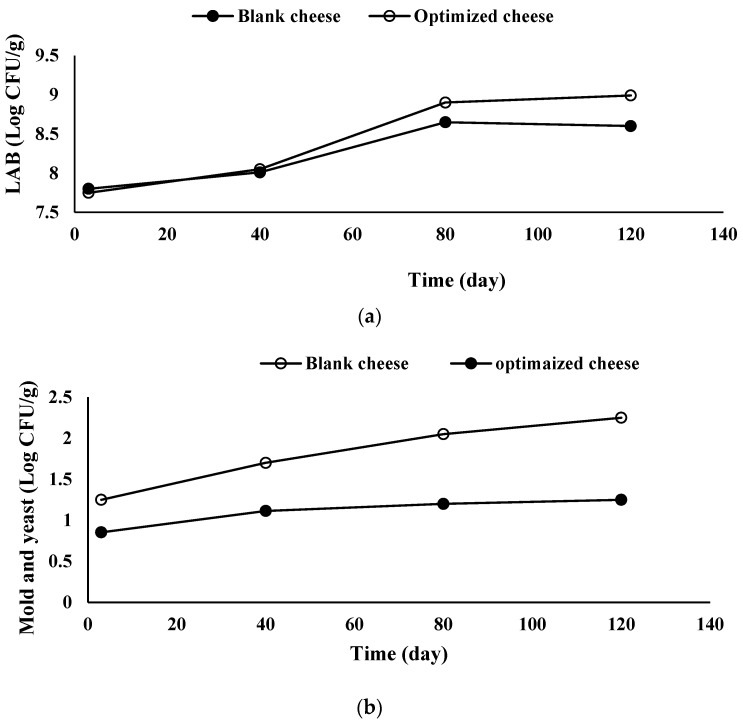
Total LAB counts (**a**) and the mold and yeast populations (**b**) of SBL-extract-added Feta-cheese (●) and control cheese (○) during 120 days of storage time.

**Table 1 foods-11-01728-t001:** The regression model coefficients for prediction of extraction yield (EY), antimicrobial activity (IZD), and antioxidant activity (DPPH) during SBL extraction at various process conditions.

Responses	Power (W)	Time (min)	Power × Time	Power^2^	Time^2^
EY (%)	1.07	1.03	0.20	−0.18	−0.25
DPPH (%)	4.40	3.66	0.67	0.90	−2.51
IZD (mm)	0.98	0.95	0.11	0.19	−0.71

**Table 2 foods-11-01728-t002:** Clustering of cheese rheological parameters.

Strength of Linkage (SI: 83.11)	Number of Linkages (SI: 86.15)	Timescale of Junction Zone (SI: 94)	Distance of Linkage (SI: 100)
*A*	*z*	*λsum*	*ξ*
*k*′	*n*′	*η*s*	
*k*″	*n*″		
	*k*″/*k*′		

*A* is the intercept of power-law model for *G** vs. *ω*, *k*′ is the intercept of power-law model for *G*′ vs. *ω*, *k*″ is the intercept of power-law model for *G*″ vs. *ω*, *z* is the network extensity parameter, *n*′ is the exponent of power-law model for *G*′ vs. *ω*, *n*″ is the exponent of power-law model for *G*″ vs. *ω*, *k*″/*k*′ is the overall loss tangent, *λsum* is the relaxation time, *η*s* is the slope of complex viscosity vs. *ω*, and *ξ* is the distance between sequential crosslinking points.

**Table 3 foods-11-01728-t003:** The RSM model coefficients of different rheological and sensorial parameters of SBL-added cheese.

Rheological Parameters	*A* (Pa·s^1/z^)	*z* (-)	*λ_rel_* (s)	*ξ* (nm)
X_1_	−4070.50	−0.69	−30.57	0.67
X_2_	1476.57	0.43	24.90	−0.38
X_3_	1318.27	−0.28	−18.51	−0.25
X_1_X_2_	2169.84	−0.40	5.81	−0.27
X_1_X_3_	2874.76	0.33	42.39	−0.28
X_2_X_3_	−2202.57	−0.07	8.51	0.23
X_1_^2^	−4211.54	−0.12	−10.21	0.36
X_2_^2^	405.78	−0.05	−8.95	0.05
X_3_^2^	1121.50	−0.67	−48.90	0.03
**Sensorial Parameters**	**Acceptance** **(-)**	**Taste** **(-)**	**Texture** **(-)**	**Odor** **(-)**
X_1_	−0.72	−0.62	−1.18	−0.76
X_2_	0.05	0.10	0.06	0.13
X_3_	−0.07	−0.20	0.09	−0.25
X_1_X_2_	-	-	0.25	-
X_1_X_3_	-	-	0.30	-
X_2_X_3_	-	-	0.18	-
X_1_^2^	-	-	−0.48	-
X_3_^2^	-	-	0.20	-

**Table 4 foods-11-01728-t004:** Changes in the physicochemical properties of SBL-extract-added UF Feta-type cheese during 60 days storage.

Time (Days)	Dry Matter (%)	Fat (%)	Ash (%)	Salt (%)	pH (-)	Syneresis (%)	TCA-SN/TN (%)	Acidity (%)	Acid Number (mEq/g)
3	32.10 ± 0.35 ^a^	12.63 ± 0.10 ^a^	2.74 ± 0.04 ^a^	1.64 ± 0.02 ^a^	4.86 ± 0.01 ^a^	-	1.63 ± 0.02 ^c^	0.89 ± 0.01 ^d^	0.10 ± 0.00 ^c^
20	32.22 ± 0.36 ^a^	12.18 ± 0.08 ^b^	2.76 ± 0.03 ^a^	1.65 ± 0.03 ^a^	4.78 ± 0.02 ^b^	-	1.66 ± 0.01 ^c^	0.94 ± 0.00 ^c^	0.14 ± 0.01 ^b^
40	32.26 ± 0.40 ^a^	12.10 ± 0.12 ^c^	2.79 ± 0.05 ^a^	1.69 ± 0.05 ^a^	4.65 ± 0.02 ^c^	-	1.78 ± 0.02 ^b^	0.99 ± 0.03 ^b^	0.16 ± 0.01 ^ab^
60	32.25 ± 0.25 ^a^	12.06 ± 0.09 ^c^	2.77 ± 0.02 ^a^	1.71 ± 0.04 ^a^	4.90 ± 0.01 ^a^	-	1.91 ± 0.03 ^a^	1.02 ± 0.02 ^a^	0.18 ± 0.00 ^a^

^a–d^: Means followed by the same lower case in the same column are not significantly different (*p* > 0.05).

**Table 5 foods-11-01728-t005:** Fatty acids composition of SBL-extract-added cheese samples at different storage times.

Fatty Acid	3rd Day	20th Day	40th Day	60th Day
C4:0 (Butyric acids)	1.70 ± 0.25 ^d^	2.45 ± 0.21 ^b^	3.09 ± 0.22 ^a^	1.82 ± 0.31 ^c^
C6:0 (caproic acid)	0.55 ± 0.08 ^d^	0.75 ± 0.07 ^c^	1.68 ± 0.09 ^a^	1.32 ± 0.10 ^b^
C8:0 (caprylic acid)	0.45 ± 0.05 ^d^	0.63 ± 0.09 ^c^	1.32 ± 0.14 ^a^	1.11 ± 0.04 ^b^
C10:0 (capric acid)	1.45 ± 0.09 ^d^	1.81 ± 0.08 ^c^	2.98 ± 0.15 ^a^	2.66 ± 0.06 ^b^
C12:0 (lauric acid)	2.19 ± 0.43 ^c^	2.33 ± 0.37 ^c^	3.68 ± 0.10 ^a^	3.21 ± 0.14 ^b^
C14:0 (myristic acid)	8.75 ± 0.12 ^d^	9.21 ± 0.23 ^c^	10.85 ± 0.20 ^a^	9.83 ± 0.40 ^b^
C16:0 (palmitic acid)	31.93 ± 0.56 ^b^	30.55 ± 0.48 ^c^	31.88 ± 1.18 ^b^	32.61 ± 0.29 ^a^
C18:0 (stearic acid)	9.90 ± 0.51 ^a^	10.10 ± 0.43 ^a^	7.21 ± 0.59 ^c^	8.51 ± 0.64 ^b^
C20:0 (arachidic)	2.81 ± 0.20 ^a^	2.71 ± 0.17 ^a^	1.41 ± 0.23 ^b^	1.40 ± 0.25 ^b^
C18:1 (oleic acid)	21.82 ± 0.75 ^a^	20.82 ± 0.64 ^a^	18.01 ± 0.20 ^c^	19.55 ± 0.30 ^b^
C18:2 (linoleic acid)	13.30 ± 0.33 ^a^	13.25 ± 0.30 ^a^	11.10 ± 0.40 ^c^	12.81 ± 0.04 ^b^
C18:3 (linolenic acid)	0.35 ± 0.05 ^c^	0.70 ± 0.08 ^a^	0.46 ± 0.06 ^b^	0.49 ± 0.03 ^b^
Short chain length fatty acid	2.81 ± 0.22 ^d^	3.73 ± 0.08 ^c^	6.09 ± 0.37 ^a^	4.14 ± 0.05 ^b^
Medium chain length fatty acid	12.30 ± 0.36 ^d^	13.33 ± 0.27 ^c^	17.51 ± 0.52 ^a^	15.67 ± 0.46 ^b^
Long chain length fatty acid	80.01 ± 0.76 ^a^	78.02 ± 0.46 ^b^	70.25 ± 1.01 ^d^	75.25 ± 0.37 ^c^

^a–d^: Means followed by the same lower case in the same row are not significantly different among different storage time (*p* > 0.05).

**Table 6 foods-11-01728-t006:** The third harmonic viscoelastic modulus to the first harmonic viscoelastic modulus ($G_{3}^{\prime }/G_{1}^{\prime }$ and $G_{3}^{\prime \prime }/G_{1}^{\prime \prime }$), strain-hardening ratio (S) and shear-thickening ratio (T) of SBL-extract-added cheese samples at different storage times.

Time (Day)	Strain (%)	S	$G_{3}^{\prime }/G_{1}^{\prime }$	T	$G_{3}^{\prime \prime }/G_{1}^{\prime \prime }$
3	0.01	^1^ 0.00 ± 0.00 ^f^	^1^ 0.01 ± 0.00 ^e^	^1^ 0.00 ± 0.00 ^f^	^1^ 0.00 ± 0.00 ^e^
	0.1	^1^ 0.01 ± 0.01 ^e^	^1^ 0.01 ± 0.00 ^d^	^1^ −0.01 ± 0.00 ^e^	^1^ 0.00 ± 0.00 ^d^
	1	^2^ 0.10 ± 0.02 ^d^	^2^ 0.09 ± 0.02 ^c^	^2^ −0.06 ± 0.02 ^d^	^2^ 0.08 ± 0.01 ^c^
	10	^3^ 0.38 ± 0.04 ^c^	^3^ 0.23 ± 0.03 ^b^	^3^ −0.20 ± 0.02 ^c^	^3^ 0.18 ± 0.02 ^b^
	100	^3^ 0.56 ± 0.06 ^b^	^3^ 0.29 ± 0.04 ^b^	^3^ −0.28 ± 0.03 ^b^	^3^ 0.20 ± 0.02 ^b^
	1000	^3^ 0.85 ± 0.05 ^a^	^3^ 0.33 ± 0.02 ^a^	^3^ −0.41 ± 0.05 ^a^	^3^ 0.25 ± 0.01 ^a^
20	0.01	^1^ 0.00 ± 0.00 ^e^	^1^ 0.00 ± 0.00 ^d^	^1^ 0.00 ± 0.00 ^d^	^1^ 0.00 ± 0.00 ^d^
	0.1	^2^ 0.06 ± 0.00 ^e^	^2^ 0.03 ± 0.00 ^d^	^2^ −0.04 ± 0.00 ^d^	^2^ 0.03 ± 0.00 ^d^
	1	^3^ 0.16 ± 0.03 ^d^	^2^ 0.12 ± 0.02 ^c^	^3^ −0.10 ± 0.02 ^c^	^2^ 0.06 ± 0.01 ^c^
	10	^4^ 0.55 ± 0.04 ^c^	^3^ 0.25 ± 0.03 ^b^	^4^ −0.30 ± 0.02 ^c^	^3^ 0.16 ± 0.02 ^b^
	100	^4^ 0.71 ± 0.06 ^b^	^3^ 0.26 ± 0.04 ^b^	^4^ −0.38 ± 0.03 ^b^	^3^ 0.18 ± 0.02 ^b^
	1000	^4^ 1.13 ± 0.05 ^a^	^3^ 0.35 ± 0.02 ^a^	^4^ −0.55 ± 0.05 ^a^	^3^ 0.23 ± 0.01 ^a^
40	0.01	^1^ 0.00 ± 0.00 ^d^	^1^ 0.00 ± 0.00 ^d^	^1^ 0.00 ± 0.00 ^d^	^1^ 0.00 ± 0.00 ^d^
	0.1	^1^ 0.01 ± 0.00 ^d^	^1^ 0.01 ± 0.00 ^d^	^1^ 0.00 ± 0.00 ^d^	^1^ 0.00 ± 0.00 ^c^
	1	^1^ 0.01 ± 0.00 ^d^	^1^ 0.01 ± 0.00 ^d^	^1^ −0.01 ± 0.00 ^d^	^1^ 0.01 ± 0.00 ^c^
	10	^2^ 0.14 ± 0.02 ^c^	^2^ 0.16 ± 0.02 ^c^	^2^ −0.14 ± 0.02 ^c^	^2^ 0.11 ± 0.03 ^b^
	100	^2^ 0.35 ± 0.02 ^b^	^2^ 0.20 ± 0.03 ^b^	^2^ −0.19 ± 0.02 ^b^	^2^ 0.14 ± 0.02 ^b^
	1000	^2^ 0.65 ± 0.07 ^a^	^2^ 0.27 ± 0.02 ^a^	^2^ −0.29 ± 0.03 ^a^	^2^ 0.19 ± 0.01 ^a^
60	0.01	^1^ 0.00 ± 0.00 ^d^	^1^ 0.00 ± 0.00 ^d^	^1^ 0.00 ± 0.00 ^d^	^1^ 0.00 ± 0.00 ^c^
	0.1	^1^ 0.01 ± 0.00 ^d^	^1^ 0.01 ± 0.00 ^d^	^1^ 0.00 ± 0.00 ^d^	^1^ 0.00 ± 0.00 ^c^
	1	^1^ 0.01 ± 0.00 ^d^	^1^ 0.01 ± 0.00 ^d^	^1^ −0.01 ± 0.00 ^d^	^1^ 0.01 ± 0.00 ^c^
	10	^1^ 0.09 ± 0.01 ^c^	^1^ 0.10 ± 0.01 ^c^	^1^ −0.11 ± 0.01 ^c^	^1^ 0.08 ± 0.01 ^b^
	100	^1^ 0.28 ± 0.01 ^b^	^1^ 0.14 ± 0.02 ^b^	^1^ −0.15 ± 0.02 ^b^	^1^ 0.10 ± 0.02 ^b^
	1000	^1^ 0.51 ± 0.06 ^a^	^1^ 0.21 ± 0.04 ^a^	^1^ −0.22 ± 0.04 ^a^	^1^ 0.15 ± 0.03 ^a^

^a–f^: Means followed by the same lower case in the same column are not significantly different among different strains at the constant storage time (*p* > 0.05). ^1–4^: Means followed by the same number in the same column are not significantly different among different storage times at the constant strain (*p* > 0.05).

**Table 7 foods-11-01728-t007:** The frequency sweep rheological parameters, textural and sensorial parameters of SBL-extract-added cheese samples at different storage times.

Frequency Sweep	3rd Day	20th Day	40th Day	60th Day
*A* (Pa·s^n^)	32,192.71 ± 1261.09 ^b^	34,775.90 ± 1167.33 ^a^	25,121.71 ± 1054.52 ^c^	18,809.36 ± 745.92 ^d^
*z* (-)	4.44 ± 0.16 ^b^	4.70 ± 0.15 ^a^	3.91 ± 0.06 ^c^	3.34 ± 0.08 ^d^
*λ_rel_* (s)	236.23 ± 9.15 ^b^	252.26 ± 8.57 ^a^	225.77 ± 4.52 ^c^	199.83 ± 7.25 ^d^
*ξ* (nm)	5.44 ± 0.19 ^c^	5.14 ± 0.20 ^d^	5.77 ± 0.09 ^b^	6.71 ± 0.24 ^a^
Textural parameters				
Hardness (N)	4.98 ± 0.05 ^b^	5.25 ± 0.08 ^a^	4.61 ± 0.08 ^c^	3.25 ± 0.12 ^d^
Cohesiveness (-)	0.40 ± 0.03 ^b^	0.49 ± 0.07 ^a^	0.35 ± 0.01 ^c^	0.30 ± 0.01 ^d^
Adhesiveness (J.S)	0.45 ± 0.05 ^c^	0.52 ± 0.06 ^c^	0.99 ± 0.01 ^b^	1.25 ± 0.01 ^a^
Springiness (mm)	4.90 ± 0.05 ^b^	5.55 ± 0.10 ^a^	3.96 ± 0.15 ^c^	2.86 ± 0.20 ^d^
Gumminess (N)	1.48 ± 0.05 ^b^	1.72 ± 0.07 ^a^	1.31 ± 0.03 ^c^	0.79 ± 0.01 ^d^
Chewiness (mm.N)	4.22 ± 0.26 ^b^	4.69 ± 0.10 ^a^	4.00 ± 0.18 ^c^	3.52 ± 0.14 ^d^
Sensorial parameters				
Acceptance	8.13 ± 0.17 ^b^	8.09 ± 0.11 ^b^	8.51 ± 0.05 ^a^	8.14 ± 0.04 ^b^
Taste	8.07 ± 0.11 ^b^	7.95 ± 0.05 ^b^	8.70 ± 0.10 ^a^	7.75 ± 0.05 ^c^
Texture	8.15 ± 0.05 ^b^	8.60 ± 0.06 ^a^	7.95 ± 0.03 ^c^	7.60 ± 0.02 ^d^
Odor	8.05 ± 0.11 ^b^	8.14 ± 0.10 ^b^	8.75 ± 0.05 ^a^	8.20 ± 0.06 ^b^
Appearance	8.15 ± 0.15 ^a^	8.10 ± 0.10 ^a^	8.06 ± 0.04 ^a^	8.08 ± 0.08 ^a^

^a–d^: Means followed by the same lower case in the same row are not significantly different (*p* > 0.05).

**Table 8 foods-11-01728-t008:** The relationship between SEM image texture indices and instrumental mechanical characteristics of UF Feta-type cheese.

Image Texture	Hardness (N)	Cohesiveness (-)	Springiness (mm)	Gumminess (N)	Chewiness (N·mm)	Adhesiveness (N·S)
Contrast	0.058	−0.006	−0.067	0.013	−0.006	0.057
Correlation	5.901	−0.816	5.236	0.035	−3.852	−9.815
Entropy	0.279	0.020	0.999	0.069	0.012	−0.890
R^2^	0.962	0.971	0.931	0.964	0.943	0.949
RMSE	0.631	0.231	1.083	0.852	1.103	1.232

**Table 9 foods-11-01728-t009:** Semicircular shape model parameters determined for the LAB counts vs. time and molds and yeast counts vs. time profiles of control and SBL-added cheeses.

Microbial Population	Parameters	Control	Treatment
Mold and Yeast	$P_{0}$ (CFU/g)	0.983 ± 0.041 ^a^	0.632 ± 0.012 ^b^
	$P_{\infty }$ (CFU/g)	2.221 ± 0.031 ^a^	1.292 ± 0.052 ^b^
	Rate (CFU/g·day)	0.011 ± 0.000 ^a^	0.005 ± 0.000 ^b^
LAB	$P_{0}$ (CFU/g)	6.502 ± 0.032 ^a^	6.453 ± 0.051 ^a^
	$P_{\infty }$ (CFU/g)	8.623 ± 0.061 ^b^	9.051 ± 0.031 ^a^
	Rate (CFU/g·day)	0.018 ± 0.000 ^b^	0.025 ± 0.001 ^a^

^a,b^: Means followed by the same lower case in the same row are not significantly different (*p* > 0.05).

## Data Availability

The authors confirm that the data supporting the findings of this study are available within the article.

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
