# Peer review of "Understanding the Relationship between Microstructure and Physicochemical Properties of Ultrafiltered Feta-Type Cheese Containing Saturea bachtiarica Leaf Extract"

_foods, 2022, doi:10.3390/foods11121728_

Round 1

Reviewer 1 Report

The manuscript describes an interesting study about the use of Saturea bachtiarica leaf extract in Ultrafiltered Feta-cheese formulation, in order to evaluate its physicochemical, rheological, textural, microstructural and sensorial properties. The manuscript shall be improved before being accepted, especially in the sessions “Introduction” and “Results”.

The introduction emphasizes the importance of developing functional foods with bioactive and nutraceutical ingredients that promotes health; however, mission objectives focus more on physicochemical, rheological, textural, microstructural and sensorial properties of the cheese added in SBL extract.  “Results” seems complete but too long, therefore sometimes difficult to understand.

The images need to be made clearer with more details in the captions and legends.

  • L33-L61: Provide a better explanation of the concept of “functional foods/nutraceutical” linking with Feta-cheese added in Saturea bachtiarica leaf extract.
  • L75: Add all the ingredients needed for the preparation of Feta-cheese
  • L80: Express a concept in a different way, without anticipating the results
  • L91: Chose the appropriate symbol to indicate the Temperature expressed in degree Celsius and use it in all the paper
  • L153: Add some more details about the control
  • L224: I think it would be better said “LAB were …”
  • L278: Add the color-legend
  • L282: Improve the link between the data in the text and Figure 1 (e.g. time 8 min)
  • L520: Improve Figure 3: letters (a,b,c,d,e,f,g,h) are not so evident
  • L523: Figure 4: indicates which images are a, b, c or d

Author Response

Dear reviewer, Allow us kindly to thank you for considering our manuscript for revision. The manuscript has been revised. Please see the attachment.

Reviewer 2 Report

Very nicely written article full of interesting results. I have only a few technical remarks.

The names of microorganisms and the Latin name of the plants should be written in italics.

In addition to the title of the chapter - Results, a discussion should be added.

Figure 3 is very poorly visible. I know it was taken from the program but still, at least the text next to the graphs should be more readable.

Author Response

(The authors gave the same response as above.)
